# Prevalence of Locomotive Organ Impairment and Associated Factors among Middle-Aged and Older People in Nan Province, Thailand

**DOI:** 10.3390/ijerph182010871

**Published:** 2021-10-15

**Authors:** Marie Niwayama, Kayako Sakisaka, Pongthep Wongwatcharapaiboon, Valika Rattanachun, Satoshi Miyata, Kenzo Takahashi

**Affiliations:** 1Graduate School of Public Health, Teikyo University, Tokyo 173-8605, Japan; sakisaka@med.teikyo-u.ac.jp (K.S.); smiyata@med.teikyo-u.ac.jp (S.M.); kenzo.takahashi.chgh@med.teikyo-u.ac.jp (K.T.); 2Nan Hospital, Nan City 55000, Nan Province, Thailand; pongthwo@gmail.com (P.W.); valeel@yahoo.com (V.R.)

**Keywords:** aging, cross-sectional study, exercise, motion, obesity, Thailand

## Abstract

While locomotive organ impairment among older people is attracting worldwide attention, this issue has not yet been widely investigated in Thailand. This study aimed to measure locomotive organ impairment prevalence and identify the determinants of locomotive function decline among middle-aged and older people in Nan Province, Thailand. This cross-sectional study included anthropometric measurements, a two-step test to investigate locomotive function, and a structured questionnaire to obtain socio-demographic and related information. Logistic regression analysis and multiple regression analysis were used to identify the determinants of locomotive organ impairment. The study participants were aged 50–87 years old (*n* = 165), and 71.5% of them had begun experiencing declining locomotive function; < 6 years of school education (adjusted odds ratio: 4.46), body mass index ≥25 kg/m^2^ (AOR: 3.06), comorbidities (AOR: 2.55), and continuous walking for <15 min (AOR: 2.51) were identified as factors associated with locomotive organ impairment. Moreover, age, knee pain, anxiety about falling in daily life, and difficulty with simple tasks were identified as factors significantly associated with exacerbated locomotive organ impairment (*p* < 0.05). Appropriate interventions such as guidance or follow-up and recommendations for exercises are needed to prevent locomotive organ impairment and improve treatment.

## 1. Introduction

According to a United Nations announcement in 2019, the proportion of the population aged ≥65 years in some Asian emerging countries is growing: Thailand (12.4%), China (11.5%), and Vietnam (7.6%) [1]. Furthermore, the World Health Organization (WHO) stated that the population aging pace is much faster than in the past and predicted that the pace would also increase greatly in low- and middle-income countries [2]. A rapidly aging society is associated with heavy social and financial burdens due to various health problems [3,4,5].

As an age-related health problem, locomotive organ impairment is beginning to attract worldwide attention. In Japan, locomotive organ impairment underlies the need for nursing care services for 21.5% of older people, and lower functional activities were associated are higher medical care costs among elderly individuals [6,7]. Thus, the Japanese Orthopaedic Association (JOA) introduced a new concept called locomotive syndrome (LS) in 2007 in order to bring attention to the need for primary prevention of locomotive organ impairment [3,6,8]. LS is a condition that necessitates or has a high risk of needing nursing care due to disorders of the locomotor organs, which consist of three main elements: bones, which give the body a framework; joints and intervertebral discs, which enable the body to be mobile; and muscles and nervous system, which move the body and/or regulate its motion [6]. Locomotive organ impairment is evaluated by simple physical tests such as the two-step test or a stand-up test, or structured questions such as the 25-question Geriatric Locomotive Function Scale (GLFS-25); these tests have been validated [9,10,11]. Such locomotor organ weakness causes mobility difficulties, including the ability to stand, walk, run, climb stairs, and perform other physical functions essential to daily life [9]. Therefore, LS is listed as one cause of shortened healthy life expectancy [10].

In Thailand, the population of older people has increased sharply. The average life expectancy of Thai people was 76.9 years in 2018 (men 73.2 years, women 80.7 years) [12]. The World Bank predicted that the population aged ≥65 in Thailand would be approximately 25% by 2040, similar to the 26.3% in Japan as of 2015 [13]. To address an aging society, the Thai government has established the Second National Plan for Older Persons (2002–2021), which focuses on developing policies and programs to support older people. The plan includes the establishment of a home-visit long-term care system and elderly centers or clubs to provide a wide range of health promotion activities; the elderly clubs were established in almost all sub-districts in Thailand, and medical care centers in communities and elderly clubs coordinated to arrange mobile units to provide information about practicing exercise, checking physical fitness, and assessing the health of older persons in communities [14]. However, the investigation focusing on locomotive organs among older people is limited in Thailand. Moreover, there have been no reports on the causes of locomotive organ impairment among the elderly in Thailand.

This study aimed to measure the prevalence of locomotive organ impairment and identify the determinants of motor function decline among middle-aged and older people in rural Thailand in order to address the maintenance of locomotive function.

## 2. Materials and Methods

### 2.1. Study Design, Study Site, Target Population

This was a cross-sectional study conducted in Nan Province in northern Thailand. Nan Province is located 660 km north of Bangkok, the capital of Thailand, and is approximately 11,000 km^2^ in size. Nan Province is in a mountainous area; the northeast region of the province adjoins the border of Laos. In 2018, the population was approximately 480 thousand, of which nearly 20% were over 65 years old, and the main industries were agriculture and tourism [15,16]. This study’s target populations were local male and female residents of Nan aged ≥50 years. The exclusion criteria were individuals who were unable to walk without support, had hemiplegia or difficulty reading and writing, or did not provide informed consent. We conducted a self-administered structured questionnaire. We also conducted anthropometric measurements, and a physical function examination called the two-step test. The staff measured participants’ height and weight using a normal height meter and bathroom scale for the anthropometric measurements, and calculated body mass index (BMI) using calculators. The study was approved by the ethical committee of the Teikyo University (approval number 19-265, 2020) in Japan, and the Nan Hospital Research Ethics Committee (approval number COA No.061, 2020) in Thailand. This study was carried out in accordance with the principles of the Declaration of Helsinki. Furthermore, an explanation of informed consent and a signature line was written on the questionnaire cover.

### 2.2. Data Collection

The survey was conducted in three areas within a 30-min drive of the center of Nan Province. These areas were the meeting place in Numlum Village, the clinic at Bupharam Village, and the clinic at Ban Maung Jung Village. Field investigators consisted of a doctor, two nurses, two health officers who had bachelor degrees in public health, and six village health volunteers. We recognized that the village health volunteers were reliable and qualified as investigators because they played important roles in public health activities in Thailand [17]. The study was conducted in August and September 2020. We applied field-based cluster sampling, in which individuals were randomly selected in the study site. We first tried to invite all eligible villagers; then, village health volunteers recruited participants for this study by making several calls to the villagers.

### 2.3. Sampling Method

We calculated the required sample size for this study using the following formula for interval estimation of population ratio [18]:n ≥ (Z_α/2_)^2^ p (1 − p)/δ^2^(1)
where *n* = sample size, δ = tolerable error, α = confidence level, *p* = population ratio. Based on this formula, the calculated sample size was 385, with α, δ, and *p* set to 95%, 5%, and 50%, respectively. We applied cluster sampling methods in the target study area. We randomly selected three villages, as shown in the Data Collection section.

### 2.4. The Two-Step Test

Regarding the evaluation of locomotive organ impairment, we adopted a two-step test (Figure 1). The two-step test was as follows [8,10,11]: (ⅰ) An individual took two steps as far as possible without losing balance, then stopped with both feet aligned. (ⅱ) The length of the two steps was defined from the starting line to the position of the tips of their toes where they stopped. (ⅲ) The length (cm) of the two steps divided by the height (cm) of the individual produced the two-step score. A two-step score ≥1.1 and <1.3 was defined as the onset of mobility function decline, ≥0.9 and <1.1 was defined as a progressive decline of mobility function, and <0.9 was when social participation was hindered due to decline of mobility function. Individuals with a two-step score <1.1 were recommended for medical treatment [10]. A two-step test was conducted in this study since it is a simple test using only a tape measure and could be used to investigate walking ability, including muscle strength, balance, and lower limbs flexibility [9].

### 2.5. Questionnaire Survey

We carried out a self-administered questionnaire, which included information on basic socioeconomic status (i.e., age, gender, marital status, number of family members, years of schooling, economic status, current job, and comorbidities), living condition (i.e., presence of low back/knee/hip pain, anxiety about falling in daily life, anxiety about difficulty with walking in the future, difficulty in putting on trousers, and difficulty with simple or load-bearing tasks), and physical activities (i.e., difficulty in walking briskly, sitting time, exercising time in a day, and continuous walking time). The contents of the questionnaire were based, in part, on previous studies [19,20].

### 2.6. Statistical Analyses

The unpaired Student’s *t*-test was used to compare continuous variables since our data were normally distributed by checking the Shapiro–Wilk test between the groups with two-step scores <1.3 and ≥1.3. Conversely, Welch’s *t*-test was used for continuous variables that were not normally distributed. The chi-square and Wilcoxon rank-sum tests were used to compare categorical variables between the two groups. Moreover, logistic regression analysis was used to identify the factors of the low two-step score. The adjusted odds ratio was calculated using a binary logistic regression analysis adjusted for age, with the presence of a two-step score <1.3 as the dependent variable and each question in the questionnaire as the independent variable. The reference groups for each associated factor were based on previous studies [19,20,21,22,23]. Furthermore, to evaluate the factors related to a two-step score decrease, multiple regression analysis using the stepwise method was performed, with the two-step score (<1.3) as the dependent variable and each question in the questionnaire as an independent variable. For variable selection in the stepwise method, the variable capture standard was taken to be at the 0.20 significance level, while the removal standard was at the 0.05 level [18]. All significance levels were set at <5%. All statistical analyses were performed using SAS 9.4 (SAS Institute, Cary, NC, USA).

## 3. Results

### 3.1. Characteristics of the Participants

A total of 165 individuals were analyzed based on the exclusion criteria among the participants in this study. The participants’ age ranged from 50–87 years, with an average age of 65.2 years (standard deviation (SD) 8.1). There were 51 male participants (30.9%), with an average age of 67.7 years (SD 6.4), and 114 female participants (69.1%), with an average age of 64.1 years (SD 8.5) (Appendix A). Overall, 118 participants (71.5%) had a two-step score <1.3. Table 1 shows the comparison between the groups with two-step scores <1.3 and ≥1.3. The average age of those with a two-step score <1.3 was significantly higher than those who scored ≥1.3 (*p* < 0.001). The years of education of those with a two-step score <1.3 were significantly lower than the other group (*p* < 0.001). Moreover, no significant difference was observed in BMI between the two groups; however, the proportion of patients with a BMI of ≥25 kg/m^2^ was 41.5% in the group with a two-step score <1.3, which was significantly higher than the other group (*p* = 0.023). As shown in Table 1, the proportion of difficulty walking briskly in the group with a two-step score <1.3 was significantly higher (*p* = 0.005). Similarly, the proportion of difficulty of load-bearing tasks in the group with a two-step score <1.3 was higher than for the other group (*p* = 0.031). Furthermore, in the group with a two-step score <1.3, 39.7% of respondents could walk continuously for ≥15 min, while 66.0% had a two-step score ≥1.3 (*p* < 0.001) (Table 1).

### 3.2. Factors Associated with a Low Two-Step Score

As shown in Table 2, logistic regression analysis adjusted for age identified that <6 years of school education (adjusted odds ratio (AOR): 4.46, 95% confidence interval (95% CI):1.77–11.28, *p* = 0.002), BMI ≥25 kg/m^2^ (AOR: 3.06, 95% CI: 1.32–7.09, *p* = 0.009), comorbidities (AOR: 2.55, 95% CI: 1.19–5.48, *p* = 0.017), and continuous walking for <15 min (AOR: 2.51, 95% CI: 1.18–5.31, *p* = 0.017) were determinants of possible locomotive organ impairment.

### 3.3. Determinants That Exacerbate the Low Two-Step Score

In the group with two-step scores <1.3, multiple regression analysis identified age (partial regression coefficient (β): −0.01, 95% CI:−0.01–0.00, *p* < 0.001), knee pain (β:−0.03, 95% CI:−0.06–0.00, *p* = 0.048), anxiety about falling in daily life (β:−0.02, 95% CI: −0.04–0.00, *p* = 0.026), and difficulty with simple tasks (β:−0.06, 95% CI:−0.10– −0.02, *p* = 0.007) as determinants of exacerbation of locomotive organ impairment (Table 3).

## 4. Discussion

Our study clarified the high prevalence (>70%) of locomotive organ impairment among middle-aged and older people in rural areas in Thailand. Moreover, this study showed that ‘shorter education period,’ ‘obesity,’ ‘comorbidities,’ and ‘short continuous walking time’ were associated with locomotive function decline.

### 4.1. Prevalence of Locomotive Organ Impairment

This study indicated that the prevalence of a two-step score <1.3, corresponding to LS, was 71.5% (men, 72.6%; women, 71.1%). In previous studies in Japan and Brazil, the prevalence of LS varied widely from 10% to 60%, although most of the previous survey results used only self-administered questionnaires in specific regional populations [21,22,23,24,25,26,27,28]. However, in a large-scale cohort study of residents conducted in urban, mountainous, and coastal areas of Japan, a two-step test, stand-up test, and the GLFS-25 were all performed, and as a result, the prevalence of LS was reported to be 69.8% [26]. Our results are close to the prevalence in the previous cohort study. In addition, the age-specific prevalence for a two-step score <1.3 was 52.5% in the 50s, 66.2% in the 60s, and 90.7% in the 70s. This suggests that the prevalence increases with age and increases remarkably in the >70 years. Similarly, a previous study in Japan reported that the spline curves of the two-step score and age showed a gradual age-dependent decrease among young and middle-aged individuals and accelerated increase among the elderly [29]. Overall, it is suggested that measures to prevent locomotive organ impairment and improve treatment are needed from an early stage. As seen in Japan, locomotive function declined in Nan Province both in the old and middle-aged, and this decline accelerated with age.

### 4.2. Locomotive Organ Impairment and Associated Factors

The results of this study also indicate that ‘shorter education period,’ ‘obesity,’ ‘comorbidities,’ and ‘short continuous walking time’ are factors significantly associated with locomotive organ impairment, while previous studies have clarified associations between gender, age, and metabolic syndrome as factors of LS and locomotive organ impairment [22,29,30]. A previous study in Japan has reported that there was no significant association between education and LS [21]. However, until the 1990s, the rate of enrolment in secondary education was only 37.19% in Thailand, as enlightenment on secondary education nationwide in Thailand was not promoted. Consequently, compulsory education for nine years was initiated in 1999 [31,32]. Furthermore, there are many previous studies on the relationship between education level and health, including high health literacy and high final education [33], and low health literacy and shorter education [34]. It is thought that the more years of education, the higher the awareness of health, the more information gathered about health, and the more people tend towards healthy behavior. Thus, our findings suggest that a lack of school education might be an important factor in locomotive organ impairment.

Obesity has become a major problem in recent years throughout Thailand. According to the WHO, the proportion of BMI ≥25 kg/m^2^ age-adjusted in Thailand in 2016 was 32.6% (men, 29.2%; women, 35.6%); it was the second-highest percentage in Southeast Asian countries, following Malaysia [35]. Therefore, the Thai government has regarded tackling obesity as an urgent issue and has taken measures, including the introduction of a sugar tax on sugar-containing beverages in 2017 [36]. Concerning the association between obesity and LS, Yoshimura et al. investigated the association between LS and a BMI ≥25 kg/m^2^ in Japan and reported that the odds ratio adjusted for age and gender was 1.344 (*p* = 0.027), which was significant [23]. Moreover, Tanaka et al. reported that LS and abdominal circumference, an indicator of visceral fat obesity, were significantly associated in both men and women [37]. Thus, obesity is strongly associated with lifestyle-related diseases and LS. Our results showed that the proportion of patients with a BMI ≥25 kg/m^2^ was 35.8% (men, 39.2%; women, 34.2%), which is similar to the proportion in all of Thailand, and that a BMI ≥25 kg/m^2^ is associated with locomotive organ impairment. These results suggest that it is necessary to prevent and improve locomotive organ impairment, obesity, and lifestyle-related diseases in Thailand.

### 4.3. Physical Activities and Locomotive Organ Impairment

According to the subgroup analysis in the group with a two-step score <1.3, ‘age,’ ‘knee pain,’ ‘anxiety about falls in daily life,’ and ‘difficulty with simple tasks’ have been shown to be significant factors that exacerbate the two-step score. It is suggested that these factors cause further motor function decline. In particular, ‘anxiety about falls in daily life’ and ‘difficulty with simple tasks’ indicate a strong decline in daily living abilities. In Japan it is therefore presumed that it is necessary to consult a doctor or have rehabilitation interventions for LS when severe locomotive organ impairment is observed [10,11]. Regarding rehabilitation interventions, Aoki et al. reported that the motor functions of the participants were significantly improved, and their adherence was high after receiving guidance on exercise therapy for LS and exercising on their own for three months while making telephone contact with experts three times a week [38]. Hashimoto et al. reported that improved locomotive function and increased frequency of telephone contact affected the continuation of the participants in a survey similar to that by Aoki et al. [39]. In addition, Nakamura et al. stated the importance of exercising safely and carefully because many middle-aged and older people had chronic degeneration of intervertebral discs and lower limb cartilages such as in the knee joint [11]. These prior studies suggest that individuals at high risk of locomotive organ impairment need interventions, such as guidance or follow-up, and recommendations for exercises. In Thailand, including Nan Province, we suggest increasing the number of physiotherapists, personnel who can provide rehabilitation guidance, and taking national counter measures such as legislation for such purposes.

This study had some limitations. First, this was a cross-sectional study. Therefore, a causal relationship could not be deduced. Second, selection bias might be considered because the participants spontaneously joined in this study; they might have high health consciousness or health concerns. It was considered that there might be uncertainty in the data provided by the self-administered questionnaire; therefore, we added real anthropometric measurement, blood pressure tests, and the two-step test in addition to the self-administered questionnaire. Third, because of the coronavirus disease pandemic, the number of participants in this survey was much lower than we planned; therefore, of the calculated 385 sample size, only 165 of 169 enrolled individuals were analyzed. Nevertheless, since the empirical power observed was ‘age’ 0.996, ‘BMI (kg/m^2^) <25’ 0.740, and ‘year of schooling’ 0.989, we considered the results of this study to be reliable. Finally, this study was conducted in one region; therefore, the results cannot be generalized sufficiently. However, despite some limitations, we clarified the actual condition of locomotive organ impairment for middle-aged and older people living in rural Thailand, where the population is aging. Our study was the first field-based study to identify the prevalence of locomotive organ impairment through actual measurement in this site. Moreover, the result of our study may be positioned as the basic evidence to promote provincial health planning to support the health of the elderly, which is a crucial issue in Thailand. Future work should be carried out in other areas in Thailand, including cities such as Bangkok, and the age group in the surveys should be expanded. Moreover, we recommend more active discussion on the actual condition of locomotive organ impairment and associated factors throughout Thailand in order to increase the number of healthy older people even in an aging society.

## 5. Conclusions

We found that the prevalence of locomotive organ impairment among residents aged ≥50 years in rural Thailand was approximately 70%. Moreover, we found that obesity, which is a recent and growing issue in Thailand, was associated with locomotive organ impairment. It is necessary to prevent and improve locomotive organ impairment from an early stage, along with obesity and lifestyle-related diseases. Future work should be carried out in other areas in Thailand, and the age group in the surveys should be expanded.

## Figures and Tables

**Figure 1 ijerph-18-10871-f001:**
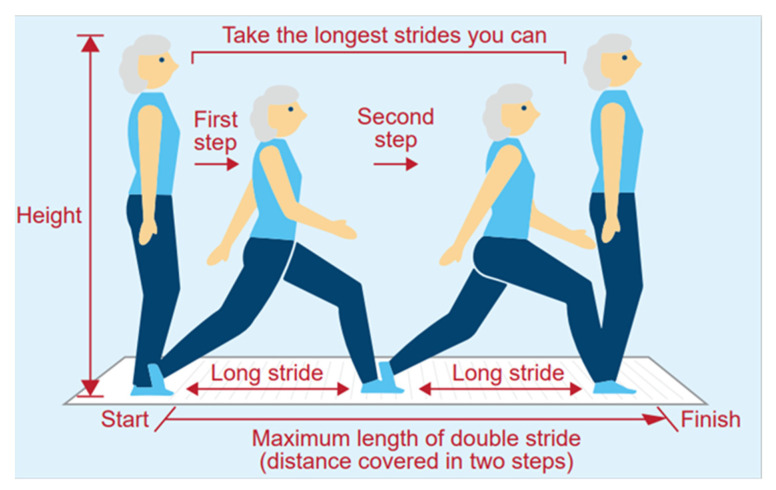
Two-step test [10].

**Table 1 ijerph-18-10871-t001:** Two-step test score and associated factors (*n* = 165).

		Two-Step Score <1.3 (*n* = 118)			Two-Step Score ≥1.3 (*n* = 47)				
		*n*	%	Mean ± SD	*n*	%	Mean ± SD	*p*-Value	
**Age**				66.9 ± 7.8			61.0 ± 7.1	<0.001 ^(a)^	***
**Sex**	Male	37	31.4		14	29.8		0.992 ^(b)^	
	Female	81	68.6		33	70.2			
**Years of schooling (year)**	Median (25th,75th)			4.0 (4.0,6.5)			9.0 (4.0,13.0)	<0.001 ^(d)^	***
**BMI (kg/m^2^)**				24.3 ± 3.7			23.3 ± 2.4	0.057 ^(c)^	
	<25.0		69	58.5	37	78.7		0.023 ^(b)^	*
	≥25.0		49	41.5	10	21.3			
**The two-step score**				1.1 ± 0.1			1.4 ± 0.1	<0.001 ^(d)^	***
**Difficulty with simple tasks (*n* = 164)**	Extremely difficult	0	0.0		0	0.0		0.300 ^(d)^	
	Considerably difficult	1	0.9		0	0.0			
	Moderately difficult	6	5.1		0	0.0			
	Mildly difficult	15	12.8		6	12.8			
	Not difficult	95	81.2		41	87.2			
**Difficulty with load-bring tasks**	Extremely difficult	11	9.3		2	4.3		0.031 ^(d)^	*
	Considerably difficult	15	12.7		4	8.5			
	Moderately difficult	29	24.6		9	19.2			
	Mildly difficult	34	28.8		13	27.7			
	Not difficult	29	24.6		19	40.4			
**Difficulty walking briskly**	Extremely difficult	5	4.2		0	0.0		0.005 ^(d)^	**
	Considerably difficult	1	0.9		0	0.0			
	Moderately difficult	15	12.7		0	0.0			
	Mildly difficult	33	28.0		12	25.5			
	Not difficult	64	54.2		35	74.5			
**Continuous walking time (*n* = 163)**	Less than 10 s	4	3.5		0	0.0		<0.001 ^(d)^	***
	About 1 min	5	4.3		0	0.0			
	About 5 min	20	17.2		2	4.3			
	About 10 min	41	35.3		14	29.8			
	15 min or more	46	39.7		31	66.0			

^(a)^ Unpaired student’s *t* test, ^(b)^ Chi-square test, ^(c)^ Welch’s *t* test, ^(d)^ Wilcoxon rank-sum test; * *p* < 0.05, ** *p* < 0.01, *** *p* < 0.001; BMI, body mass index; SD, standard deviation. Comparing the groups with two-step scores <1.3 and ≥1.3, significant differences were observed in average age, years of education, BMI of ≥ 25 kg/m^2^, difficulty with load-bring tasks, difficulty in walking briskly, and continuous walking time.

**Table 2 ijerph-18-10871-t002:** Factors associated with the low two-step score (<1.3) (*n* = 165).

			COR	95% CI		*p*-Value		AOR	95%CI		*p*-Value	
**Age (years)**	50–59		1.00									
	60–69		2.09	0.94	4.66	0.071						
	≥70		10.40	3.44	31.48	<0.001	***					
**Sex**	Male		1.00					1.00				
	Female		0.93	0.45	1.94	0.844		1.57	0.68	3.65	0.291	
**Years of schooling**	≥12 years		1.00					1.00				
	7–11 years		2.33	0.78	6.92	0.129		2.61	0.81	8.43	0.109	
	0–6 years		5.69	2.38	13.58	<0.001	***	4.46	1.77	11.28	0.002	**
**Comorbidities**	Hypertension	-	1.00					1.00				
		+	1.87	0.94	3.72	0.076		1.62	0.76	3.44	0.210	
	Hyperlipidemia	-	1.00					1.00				
		+	2.69	1.22	5.92	0.014	*	2.14	0.93	4.94	0.074	
	Diabetes	-	1.00					1.00				
		+	2.70	0.98	7.49	0.056		2.31	0.79	6.70	0.125	
	No underlying diseases		1.00					1.00				
	Underlying diseases		3.18	1.55	6.51	0.002	**	2.55	1.19	5.48	0.017	*
**BMI (kg/m^2^)**	18.5–24.9		1.00					1.00				
	<18.5		2.81	0.32	25.02	0.354		0.96	0.09	10.27	0.974	
	≥25.0		2.76	1.25	6.09	0.012	*	3.06	1.32	7.09	0.009	**
**Low back pain**	-		1.00					1.00				
	+		1.20	0.60	2.40	0.602		1.19	0.57	2.48	0.653	
**Knee pain**	-		1.00					1.00				
	+		1.95	0.97	3.92	0.059		1.91	0.91	4.02	0.087	
**Anxious about fall in daily life**	No		1.00					1.00				
	Yes		1.63	0.82	3.21	0.161		1.41	0.68	2.92	0.362	
**Anxious about difficulty with walking in the future**	No		1.00					1.00				
	Yes		0.81	0.41	1.59	0.538		0.75	0.36	1.57	0.444	
**Difficulty in putting on trousers**	Not difficult		1.00					1.00				
	Difficult		1.33	0.61	2.90	0.482		0.96	0.41	2.25	0.924	
**Difficulty walking briskly**	Not difficult		1.00					1.00				
	Difficult		2.46	1.16	5.21	0.019	*	1.72	0.77	3.82	0.187	
**Difficulty with simple tasks**	Not difficult		1.00					1.00				
	Difficult		1.58	0.60	4.19	0.356		1.49	0.51	4.38	0.473	
**Difficulty with load-bearing tasks**	Not difficult		1.00					1.00				
	Difficult		2.08	1.02	4.27	0.045	*	1.83	0.85	3.95	0.122	
**Weekday sitting time**	<360 min		1.00					1.00				
	≥360 min		1.69	0.53	5.34	0.374		1.69	0.50	5.73	0.402	
**Weekend sitting time**	<360 min		1.00					1.00				
	≥360 min		1.32	0.45	3.83	0.612		1.13	0.36	3.55	0.829	
**Exercising time in a day**	≥30 min		1.00					1.00				
	<30 min		1.85	0.81	4.23	0.143		1.40	0.58	3.40	0.452	
**Continuous walking time**	≥15 min		1.00					1.00				
	<15 min		2.95	1.45	5.99	0.003	**	2.51	1.18	5.31	0.017	*

COR, crude odds ratio; AOR, adjusted odds ratio; CI, confidence interval, adjusted for age; BMI, body mass index; * *p* < 0.05, ** *p* < 0.01, *** *p* < 0.001, + positive, – negative. Six years of school education, comorbidities, and continuous walking <15 min were identified as factors associated with locomotive organ impairment in logistic regression analysis.

**Table 3 ijerph-18-10871-t003:** Determinants of low two-step test scores (<1.3) (*n* = 118).

	Partial Regression Coefficient	95% CI		*p*-Value	
**Age (years)**	−0.01	−0.01	0.00	<0.001	***
**Knee pain**	−0.03	−0.06	0.00	0.048	*
**Anxiety about falling in daily life**	−0.02	−0.04	0.00	0.026	*
**Difficulty with simple tasks**	−0.06	−0.10	−0.02	0.007	**

* *p* < 0.05, ** *p* < 0.01, *** *p* < 0.001; Multiple regression analysis using the stepwise method was performed with the two-step score as the dependent variable. Age Number of family members, Year of schooling, Economic status, Number of comorbidities, Low back pain, Knee pain, Hip pain, Anxiety about falling in daily life, Anxiety about difficulty with walking in the future, Difficulty in putting on trousers, Difficulty in walking briskly, Difficulty with simple tasks, Difficulty with load-bearing tasks, Weekday sitting time, Weekend sitting time, Exercise time in a day, Continuous walking time, and BMI were independent variables. The variable capture standard was 0.20, removal standard was 0.05. Age, knee pain, anxiety about falling in daily life, and difficulty with simple tasks were identified as factors that exacerbated declining locomotive function in multiple regression analysis.

## Data Availability

The datasets used and analyzed during this study are available from the corresponding authors upon reasonable request.

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
