# Peer review of "Prevalence of Locomotive Organ Impairment and Associated Factors among Middle-Aged and Older People in Nan Province, Thailand"

_ijerph, 2021, doi:10.3390/ijerph182010871_

Round 1
Reviewer 1 Report
It is a necessary study, but I find enough questions that have to be clarified to see if it has enough quality to be published, especially in the statistics section. The English language needs to be improved, it is understood, but it has numerous writing errors and spelling. It also has important biases such as the number of participants that was less than expected.
Abstract
You must add the number of study subjects and their main sociodemographic characteristics in this section.
Keywords
Check if it is in the thesaurus or pubmed as MeSH
Introduction
Put more references on locomotor impairment, if the relationship with any variable has been previously studied, Talk about more validated methods for measuring locomotor impairment, is there only the two step test? Is this validated? Talk about what it measures and how those qualities influence the musculoskeletal system.
State the hypothesis in the introduction, after the purpose of the study.
Material and methods
Exclusion criteria I feel that more criteria are lacking, such as recent trauma, surgeries and medication.
Define the variables.
No software was used for the calculation of the sample? Give justification for the reference of the formula for the calculation of the sample size. If the number of participants was less than expected, then calculate the power and error B for the number of subjects that were successfully recruited.
Statistics. It is not clear if they used normality tests of the variables, the reason for using the welch t test, if they have already used the student t test.
And the Mann–Whitney U tests is not for categorical variables. It is a non-parametric test but of continuous variables.
Results
Tables, its legend is not very explanatory.
Author Response
Thank you very much for the thoughtful and constructive feedback.
Please see the attachment.

Reviewer 2 Report
Dear Authors,
Thanks for invited me to review this interesting study entitled: “Prevalence of locomotive organ impairment and associated factors among middle-aged and older people in Nan Province, Thailand.” I want to congratulate the authors for their hard works. This manuscript has generally performed in a good way. However, the reviewer might have missed some points from the authors or given some advice to them. The comments and suggestions have listed as follows:
Abstract
Introduction: The story could be improved to fill the gaps among the points you wish to say. Some parts of this section might be too stiff for the reviewer.
- Lines 30-45: It contains many things in one paragraph and makes it hard to read (or could miss the point that you wish to say). It is recommended to perform one specific topic in one paragraph deeply and leading the next paragraph (next topic you want to say).
- Lines 33-35: the authors mentioned that “80% of older population would be living in the low- and middle-income countries by 2050.” This sentence is not clear to the reviewer and may be misunderstood by the readers. It sounds like the older people will be living in the low- and middle-income countries because of their age. The authors should clarify the statement.
- Lines 38-45: The authors provided the definition of LS and some studies from Japan, which is good. But why only the Japanese cases? How about the other countries? This question may be related to lines 30-31; at the beginning of this manuscript, you MAINLY mentioned the current aging status of Japan instead of Thailand. It reveals an unbalance in the first look for the study.
- Lines 49-51: The authors mentioned the Second National Plan for Older Persons (2002-2021). But…how the Plan goes? Is there any achievement or improvement for the older people in Thailand (or maybe not helpful)? What are the differences between this plan and your study? These should be addressed explicitly for your research purpose.
- The entire section provided only one message for the importance of locomotive organ impairment but rarely mentioned how it happens and what factors could cost it to happen. The authors should provide more information related to their study topic.
Methods: To be easier for the reader, the reviewer suggests the author re-arrange the sections' order. For example, you may provide the information of “Study design, study site, target population” at first, then “Data collection” and “sampling method” (I suggest combining them in one section). After that, you can present the measurements and test protocol. Finally, the statistic approaches the last.
- Lines 80-92: Is there any report on the reliability and validity of the two-step test?
- Lines 94-100: Same as the step test, the reliability and validity reports are suggested to be provided.
- Lines 102-106: The sample size calculation is one of the works in sampling. However, in the reviewer’s opinion, the authors should provide the sampling “design” of this study (e.g., randomized vs. purposed sampling?). Moreover, look at the calculated sample size in line 105 and shows 385, but the result in line 134 was 165. That makes your sampling result insufficient, isn’t it? The authors must clarify such unbalancing.
- Lines 108-116: The section is highly recommended to be re-organized, and some important information might be missing. For example, the qualification or credibility of the field investigators (especially the village volunteers), the environment for data collection, how many invitations your sent (called)…etc. Also, the review suggests the authors mention the volunteers' role right after you mentioned them, instead of the last.
- Lines 118-131: The authors previously mentioned several cutoff points for the two-step test, but only a ratio of 1.3 is applied for population dichotomization. Can the authors provide the practical approaches and their reason/reference on data preprocessing?
- Lines 127-129: the authors mentioned: “For variable selection in the stepwise method, the variable capture standard was taken to be at the 128 0.20 significance level.” Please provide the reference.
- Lines 122-123: Why the authors only used age model adjustment? Many more factors may confound the results of analysis, as the t-test/chi-square test results provided. Alternatively, at least considered the gender differences in the analysis.
8.
Results: In general, the results section looks fine. But:
- The authors should mention or provide the reference on selecting the reference group for each associated factor.
- Also, it seems the model adjustment has used more than one confounder, as you mentioned in line 156: “After controlling for possible confounding FACTORS.” Then the authors should clarify what confounding factors you have used and re-organize the statement in the statistical analysis section.
- Furthermore, the cutoffs for each associated factor seem not provided in the manuscript. Please clarify the missing.
Discussion and conclusion: performed reasonably.
- Lines 263-265: It seems the authors are trying to connect the LS prevalence between Thailand and Japan in the manuscript. However, the reason has not been talked about much. If there is a reason or idea, the authors should explain it clearly for the readers. It is kind of confusing.
- Line 271: the authors concluded: “…was approximately 70%, and it increased with age.” It may imply that locomotive organ impairment is changing over time. However, as mentioned in the limitation, no cause-and-effect relationship can be guaranteed in this study. The authors should carefully use the words such as increase, decrease, improve…etc. throughout the entire manuscript.
Author Response

(The authors gave the same response as above.)

Reviewer 3 Report
Dear authors "Prevalence of locomotive organ impairment and associated factors among middle-aged and older people in Nan Province, Thailand",
This was an observational study that aimed to report the prevalence of locomotive organ impairment and to identify determinants of motor function decline in young and older Thai adults. The organization of the manuscript is fine but it can be improved if reported according to the STROBE guidelines. Authors should also report some crucial methodological information and provide some graphics to understand the data distribution. English needs to be improved throughout, the manuscript would benefit from a thorough proofread by a native English speaker to improve grammar.
Specific comments:
Line 11-12: Please merge the sentences (grammar editing).
Lines 12-13: "to clarify locomotive organ impairment prevalence"? Please change the verb since it does not represent a valid study aim.
Line 15: Authors mentioned "anthropometric measurements" but only body mass and stature were measured. Please modify since several other anthropometry-based calculations might have been included to strengthen the analysis (this reflects a flaw in the study design).
Line 15-17: Instead of the description of the test (The
two-step test investigates comprehensive walking ability and is used as an index of the locomotive syndrome, declining mobility due to locomotive organ impairment.) you might include information about the participants, methods, and statistical approach.
Line 23-25: Authors might improve the conclusion statement. Only "for those at high risks"?
Line 26-27: Authors are requested to use MeSH terms. None of those are valid. Why "rehabilitation"?
Line 56-58: "to clarify locomotive organ impairment prevalence"? Please change the verb since it does not represent a valid study aim.
Line 60: Authors are required to structure the manuscript according to the STROBE guidelines to enhance the quality and transparency of the research. Primary and secondary outcomes are missing.
Line 70: Is the GLFS-25 instrument valid for the Thailand population? Please cite accordingly. If not, please argue why did not perform piloting. A copy of this instrument as a supplementary file would be appreciated to support open science and reproducibility.
Line 73: RE, "height" with "stature". RE, "weight" with "body mass". Could the authors please explain why did not other more relevant anthropometry data (e.g., waist girth)?
Cf, the locomotive index (i.e., an indicator of biomechanical efficiency = [adipose mass + residual mass / muscle mass + bone mass]) would have been very interesting considering your sample size, study aim, and statistical approach.
Line 75: If considered, please cite the "Declaration of Helsinki" accordingly.
Line 80-92: Please report the CV or reliability of this test.
Line 93-100: Citation missing. Or did the authors develop and validate this survey instrument? Please argument.
Line 107: I congratulate the authors for developing this research effort through the COVID-19 pandemic.
Line 117: Like the statistical approach. Please mention the AOR here and try to explain it for clarification to the readers.
Line 134: I am aware you did not accomplish the sample size due to the COVID-19 pandemic. However, please report in methods that you implemented through the study, in this case, a convenience (non-probabilistic) sample should be remarked.
Line 136-137: I would like to see the data distribution. Suggest including some Seaborn stripplot with violin plot bars in front of points.
Line 152: Please format the table for better readability.
Line 172-177: The text of the figure footnote should be in the "statistical analysis" subsection.
Line 179: Discussion is well-written. I suggest highlighting the uncertainty of the data given the self-administered nature of the survey instrument.
Great that you preserved the order of your pre-defined primary and secondary outcomes; however, please remark those outcomes in the methods section accordingly.
Line 257-259: You can include here the initial aim to get a calculated sample size.
Line 271: 70% for both middle-age and older population?
Grammar editing.
Author Response

(The authors gave the same response as above.)

Reviewer 4 Report
This manuscript entitled “Prevalence of locomotive organ impairment and associated factors among middle-aged and older people in Nan Province, Thailand” primarily aimed to analyses the determinants of locomotive function decline among middle-aged and older people in Thailand. The results of this study might provide guidance for public health and health care. While it is an interesting topic, there are several key flaws or questions must be addressed, which are listed in the specific comments below.
Specific comments
- In the abstract part, the author provided too much background descriptions in this part, which may be too long-winded. I suggest that the authors provide more detailed descriptions of the methods, results, and conclusions of this study in this part.
- How many participants were included in this study? Please provide their relevant information.
- “This cross-sectional study included anthropometric measurements, structured questionnaires, and a two-step test.” How is demographic information measured? What equipment is used? Please provide more detailed descriptions. (Line 14-15)
- In the introduction part. “A rapidly aging society is associated with a heavy burden of various health problems among senior citizens.” Please provide more than one references to prove this statement. (Line 35-36)
- “In Japan, the Japanese Orthopaedic Association (JOA) emphasized a new concept called locomotive syndrome (LS) in 2007 to bring attention to the need for primary pre-vention of locomotive organ impairment.” Please add some references to support this sentence. (Line 38-40)
- “Such locomotor organs weakness causes mobility difficulties.” Which locomotor organs? please give a few examples as well as previous research about it. (Line 42-44)
- “The Thai government has established the Second National Plan for Older Persons (2002–2021) to address an aging society, which focuses on developing policies and programs to support older people.” Please add a reference to support this sentence. (Line 49-51)
- “The Thai government has established the Second National Plan for Older Persons (2002–2021) to address an aging society, which focuses on developing policies and programs to support older people.” Please add some references to support this sentence.
- What is the author's research hypothesis, which I suggest to be added to the last paragraph of the introduction?
- In the materials and methods part. “This was a cross-sectional study conducted in Nan Province in northern Thailand. Nan Province is located 660 km north of Bangkok” Why did the author choose Nan Province as the subject area of this study. (Line 62-63)
- In the reviewer's opinion, the authors provide a lot of detailed study design and data analyses, which makes the description too lengthy. To make the process clearer to the reader, it is recommended to add a flowchart.
- “We conducted a self-administered structured questionnaire including the 25-question Geriatric Locomotive Function Scale (GLFS-25), which is widely used to evaluate LS.” This sentence is too general and unclear. Can you be more specific about what are the 25-question Geriatric Locomotive Function Scale (GLFS-25).(69-71)
- “For the anthropometric measurements, the staff measured the height and weight using measuring devices and calculated the body mass index (BMI) using calculators.” Please provide more detail information about the measuring devices. (Line 72-74)
- “Regarding the evaluation of locomotive organ impairment, we adopted a two-step test (Figure 1).” Where is the figure 1 in this manuscript? (Line 80-81)
- “A two-step test was conducted in this study since it is a simple test using only a tape measure and could be used to investigate walking ability, including muscle strength, balance, and lower limbs flexibility” The reviewer suggested that the author provide a more detailed description of this step of the measurement process. (Line 89-92)
- In the Result part. “A total of 169 individuals participated in this survey.” The reviewer suggested that the author use a form to present the subject information. (Line 134-135)
- In the discussion part. Thus, “these results suggest that it is necessary to take measures against locomotive organ impairment, obesity, and lifestyle-related diseases in Thailand.” This sentence is too general and unclear. What measures?Please provide more specific information. (Line 229-231)
- “In addition, Nakamura et al. stated importance of exercise safely and carefully” The noun phrase importance seems to be missing a determiner before it. Consider adding an article. (Line 246-248)
- The novelty and value of the study should be highlighted in this part.
- In the Conclusion part. please show detailed findings about this manuscript as well as what are the contributions for future clinical or scientific research.
Author Response

(The authors gave the same response as above.)

Round 2
Reviewer 1 Report
Dear authors. I believe that the improvements are appropriate and that it can be accepted in the present version.
Author Response
We really appreciate you taking the time to review our manuscript. The manuscript has now been edited by a professional editing service to strengthen the English linguistic component of the manuscript.
Reviewer 4 Report
Authors made substantial revisions to improve the clarity of this manuscript. However, the writing is suggested to get polished again for understanding.
Author Response
We really appreciate you taking the time to review our manuscript. The manuscript has now been edited by a professional editing service to strengthen the English linguistic component of the manuscript. Moreover, we attached the official editing certificate from our editing service to this manuscript.
